

# Rethinking the use of finite element simulations in comparative biomechanics research

Z. Jack Tseng

Department of Integrative Biology and Museum of Paleontology, University of California, Berkeley, CA, USA

## ABSTRACT

In the past 15 years, the finite element (FE) method has become a ubiquitous tool in the repertoire of evolutionary biologists. The method is used to estimate and compare biomechanical performance implicated as selective factors in the evolution of morphological structures. A feature common to many comparative studies using 3D FE simulations is small taxonomic sample sizes. The time-consuming nature of FE model construction is considered a main limiting factor in taxonomic breadth of comparative FE analyses. Using a composite FE model dataset, I show that the combination of small taxonomic sample sizes and comparative FE data in analyses of evolutionary associations of biomechanical performance to feeding ecology generates artificially elevated correlations. Such biases introduce false positives into interpretations of clade-level trends. Considering this potential pitfall, recommendations are provided to consider the ways FE analyses are best used to address both taxon-specific and clade-level evolutionary questions.

## INTRODUCTION

Structure-function relationships underlie many hypotheses about patterns of morphological disparity in organisms (*Lauder, 1981*; *Lauder & Thomason, 1995*). Among the major tools in estimating the functional performance of vertebrate structures in particular is the use of biomechanical simulations to predict traits like bite force, structural stiffness, mechanical efficiency, etc. (*Richmond et al., 2005*; *Ross, 2005*). In the past 15 years, the adaptation of finite element (FE) modeling, an engineering method for solving load-deformation scenarios using principles of continuum mechanics, to biological questions has been greatly expanded across the broad spectrum of organismal study systems. An important feature of FE approaches is the ability to quantitatively test functional morphological hypotheses, contrasting it with largely qualitative conclusions and inferences in classic comparative and functional anatomical approaches A search in the "Web of Science" database returned 768 publications between 2005 and 2020 on using FE models in the study of evolution and biology (Fig. 1A). Research using FE analysis of the vertebrate skeleton covers topics such as inferring locomotory and masticatory

Corresponding author
Z. Jack Tseng, zjt@berkeley.edu

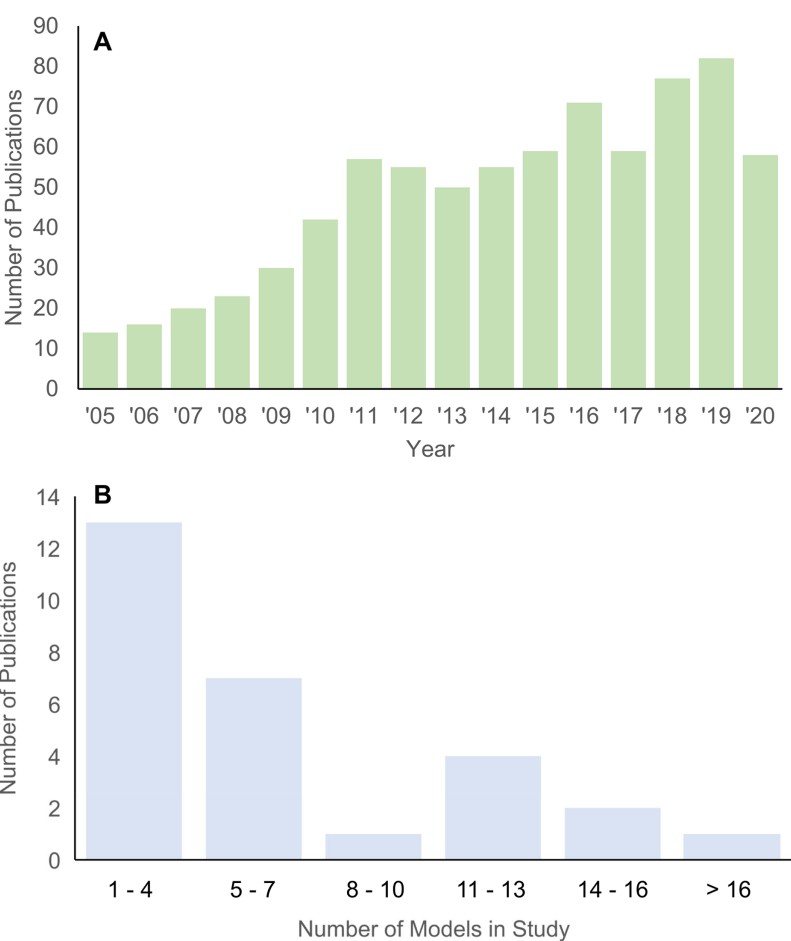

**Figure 1 Histograms of publications using FE analysis from 2005 to 2020.** (A) Total number of publications from each year in the Web of Science database (searched 22 December 2020) using the key words "finite element" + "evolution" + bio*. (B) Number of taxa included in each publication on vertebrate skull biomechanics listed in Web of Science (searched 15 December 2020) using key words "finite element" + "skull" + phylo*.

performance in the vertebrate fossil record (*Rayfield, 2007*), morphofunctional evolution in entire clades (*Pierce, Angielczyk & Rayfield, 2008*), to evolutionary optimization of functional morphological attributes (*Polly et al., 2016*), and others.

Studies using the FE method to test hypotheses about vertebrate structure-function fall into two major categories: 2D and 3D analyses. 2D FE models are typically derived from photographs of specimens, whereas 3D FE models are typically derived from computed tomography (CT) or surface scans. The main trade-off between 2D and 3D approaches is model sample size vs time investment in building each model (*Morales-García et al., 2019*). 2D models are quicker to build and allow for larger taxonomic sample sizes, but the extent of model simplification restricts its application to structures whose function can be reasonably approximated in two dimensions. 3D models can provide a fuller characterization of the morphology at hand but are much more time-consuming to build. As such, most studies using 3D FE models examine fewer than 10 taxa (Fig. 1B).

Given the steady increase in research studies employing FE methods to address comparative biomechanical and evolutionary questions, it is critical for both practitioners of FE analyses and researchers considering using the FE toolkit to test biomechanical hypotheses to be able to design studies efficiently using this time-intensive method. To highlight the important issue of sample size in 3D FE studies of structure-function relationships, here I ask the question: are time-bottlenecked small-sample 3D FE datasets adequate in reproducing performance-ecology correlations observed in broader taxonomic datasets? Given the already rich literature in using 3D FE analyses in comparative skull biomechanics research, I take a meta-analysis approach to addressing this question using mainly published studies.

The study system I use to demonstrate the effect of taxonomic sample size on performance-ecology relationships is the skull. Although the FE method is applicable to any morphological system whose geometry and biophysical boundary conditions can be digitized and parameterized, most studies in vertebrates employing FE modeling in a comparative context have done so to study skull biomechanical performance (*Ross, 2005*). Nevertheless, the effects of taxonomic sample size are expected to be shared in part by other study systems. Therefore, the findings from these analyses are expected to be relevant to researchers in comparative biology, paleobiology, bioengineering, and biomedical engineering fields that use multi-taxon comparative FE datasets to test structure-function hypotheses.

## SURVEY METHODOLOGY

I tallied the number of taxa included in published FE studies in general (Fig. 1A) and specifically of the vertebrate skull from 2005 to 2020 for 3D-model based analyses (Fig. 1B). Total number of peer-reviewed publications using FE methods were extracted from the Web of Science database (accessed 22 December 2020) using the key words "finite element" + "evolution" + bio*.

A second survey of the literature was conducted on FE studies that specifically address vertebrate skull biomechanics in a comparative context by searches in both Web of Science and Google Scholar (both accessed 15 December 2020) using key words "finite element" + "skull" + phylo*. The year of publication was constrained to between 2005 and 2020. The total number of unique species studied in each surveyed publication was counted. The surveyed publications were further vetted by removing all studies that used 2D FE models (Fig. 1B). Out of the 28 studies obtained from the skull FE survey (Table S1), the FE model construction methodology used in each survey publication was noted and a composite FE model dataset was constructed (see next section).

### Meta-analysis methodology

Based on the surveyed publication dataset, I compiled model results from *Prybyla, Tseng & Flynn (2018)*, *Pérez-Ramos et al. (2020)*, as well as three additional, new models that complement those in the two published studies to assemble a dataset of 3D cranial FE output data (here on referred to as the "full dataset"; Table 1). Data from those studies were chosen because together they represent the largest sample of FE models in the

**Table 1 Biomechanical attributes from finite element simulations used in bootstrap analyses.** ACME, adjusted canine mechanical efficiency; AP4ME, adjusted premolar four mechanical efficiency; ADJCSE, adjusted canine strain energy (in Joules); ADJP4SE, adjusted premolar four strain energy (in Joules).

| Genus | Species | ACME | AP4ME | ADJCSE | ADJP4SE | References |
|---|---|---|---|---|---|---|
| *Ailuropoda* | *melanoleuca* | 0.1688 | 0.2453 | 0.5243 | 0.4685 | *Pérez-Ramos et al. (2020)* |
| *Ailurus* | *fulgens* | 0.1664 | 0.2588 | 0.5762 | 0.5442 | *Prybyla, Tseng & Flynn (2018)* |
| *Aonyx* | *capensis* | 0.2483 | 0.3566 | 0.9297 | 1.1842 | This study |
| *Bassariscus* | *astutus* | 0.1458 | 0.2380 | 0.4134 | 0.4425 | *Prybyla, Tseng & Flynn (2018)* |
| *Canis* | *lupus* | 0.1032 | 0.1834 | 0.8375 | 0.7801 | *Prybyla, Tseng & Flynn (2018)* |
| *Canis* | *mesomelas* | 0.1507 | 0.2454 | 1.1223 | 0.9307 | *Prybyla, Tseng & Flynn (2018)* |
| *Crocuta* | *crocuta* | 0.1759 | 0.2894 | 0.4261 | 0.4750 | *Prybyla, Tseng & Flynn (2018)* |
| *Gulo* | *gulo* | 0.2585 | 0.3552 | 0.3664 | 0.3003 | *Prybyla, Tseng & Flynn (2018)* |
| *Helarctos* | *malayanus* | 0.1678 | 0.2253 | 0.5731 | 0.4959 | *Pérez-Ramos et al. (2020)* |
| *Herpestes* | *javanicus* | 0.1186 | 0.1885 | 0.6201 | 0.5672 | *Prybyla, Tseng & Flynn (2018)* |
| *Hydrictis* | *maculicollis* | 0.3042 | 0.5123 | 1.1761 | 2.4438 | This study |
| *Lutra* | *lutra* | 0.2000 | 0.3122 | 1.1488 | 1.1752 | This study |
| *Lycaon* | *pictus* | 0.2056 | 0.3179 | 1.2854 | 1.2611 | *Prybyla, Tseng & Flynn (2018)* |
| *Melursus* | *ursinus* | 0.1706 | 0.2412 | 0.6626 | 0.6374 | *Pérez-Ramos et al. (2020)* |
| *Mephitis* | *mephitis* | 0.1032 | 0.1397 | 0.8477 | 0.8436 | *Prybyla, Tseng & Flynn (2018)* |
| *Panthera* | *pardus* | 0.0850 | 0.1430 | 0.5437 | 0.4759 | *Prybyla, Tseng & Flynn (2018)* |
| *Parahyaena* | *brunnea* | 0.1724 | 0.3306 | 0.5206 | 0.6353 | *Prybyla, Tseng & Flynn (2018)* |
| *Potos* | *flavus* | 0.2637 | 0.3716 | 1.8570 | 1.2252 | *Prybyla, Tseng & Flynn (2018)* |
| *Procyon* | *lotor* | 0.1162 | 0.1634 | 0.8562 | 0.7668 | *Prybyla, Tseng & Flynn (2018)* |
| *Spilogale* | *putorius* | 0.1059 | 0.1491 | 0.2593 | 0.2534 | *Prybyla, Tseng & Flynn (2018)* |
| *Taxidea* | *taxus* | 0.3404 | 0.5395 | 0.5622 | 0.4336 | *Prybyla, Tseng & Flynn (2018)* |
| *Tremarctos* | *ornatus* | 0.1423 | 0.1914 | 0.4451 | 0.4271 | *Pérez-Ramos et al. (2020)* |
| *Urocyon* | *cinereoargenteus* | 0.1395 | 0.2243 | 1.0640 | 0.9317 | *Prybyla, Tseng & Flynn (2018)* |
| *Ursus* | *americanus* | 0.1422 | 0.2015 | 0.5321 | 0.6297 | *Pérez-Ramos et al. (2020)* |
| *Ursus* | *arctos* | 0.1647 | 0.1824 | 0.5123 | 0.4940 | *Pérez-Ramos et al. (2020)* |
| *Ursus* | *maritimus* | 0.1248 | 0.1868 | 0.4922 | 0.5513 | *Pérez-Ramos et al. (2020)* |
| *Ursus* | *thibetanus* | 0.1429 | 0.2038 | 0.5227 | 0.4759 | *Pérez-Ramos et al. (2020)* |

literature constructed using a single protocol, thus reducing confounding factors from differences in FE model construction methodology and software programs used. The full dataset included published data from FE models of 24 carnivoran species and new data from three species (*Aonyx capensis*, *Hydrictis maculicollis*, and *Lutra lutra*). The new models were built following *Pérez-Ramos et al. (2020)* protocol and summarized below. Sixteen extant species FE models out of 21 taxa from *Prybyla, Tseng & Flynn (2018)* were included; five taxa from that study were excluded (*Ursus arctos* and *U. maritimus* overlapped with the *Pérez-Ramos et al. (2020)* study, and *Leptarctus primus*, *Thinocyon velox*, and *Oodectes herpestoides* were excluded because they represent fossil taxa without ecological trait data). Likewise, 4 of 12 taxa from *Pérez-Ramos et al. (2020)* were

excluded because they represented fossil taxa (*U. ingressus*, *U. spelaeus spelaeus*, *U. spelaeus eremus*, *U. spelaeus ladinicus*).

The FE model building protocol was identical across all 27 models used in this study except for the elastic moduli values (20 GPa for models from *Prybyla, Tseng & Flynn (2018)*, 18 GPa for models from *Pérez-Ramos et al. (2020)* and this study), which were standardized using a secondary linear regression analysis. Briefly, the FE model and simulation protocol include capturing the 3D geometry of each skull specimen using CT scanning (data for the new models constructed were downloaded from scans uploaded to MorphoSource.org by *Tseng et al. (2017)*). Three-dimensional skull models were constructed from voxels selected using threshold segmentation to include all cortical bone in Avizo (Thermo Fisher Scientific, Hillsboro, OR, USA) or Dragonfly (Object Research Systems, Montreal, QC, Canada) software. All remnants of turbinate bones in the nasal cavity were removed from the 3D surface mesh in Geomagic Wrap (3D systems, Rock Hill, SC, USA), where mesh element aspect ratios are constrained to a maximum of 10, and all non-anatomical holes in the mesh patched. Solid meshes were constructed using 4-noded tetrahedral elements in Strand7 (Strand7 Pty. Ltd., Sydney, Australia). Input muscle forces were simulated using muscle forces estimates derived from a modified dry skull method (*Thomason, 1991*) for estimating muscle attachment areas of the temporalis, masseter, and medial pterygoid muscles. Force vectors mimicking muscle wrapping over the insertion sites were generated using the BoneLoad MatLab script (*Grosse et al., 2007*).

New models generated in this study used elastic moduli of 18 GPa as in *Pérez-Ramos et al. (2020)*; models from *Prybyla, Tseng & Flynn (2018)* used the slightly higher 20 GPa as their elastic moduli. As such, moduli correction was necessary to directly compare the outputs of the simulations from the different studies. I sampled 9 taxa from the *Prybyla, Tseng & Flynn (2018)* dataset and reanalyzed the models using elastic moduli of 18 GPa. The resulting deviations were then correlated using linear regression analyses to obtain correction factors for 20 GPa-based data to their 18 GPa equivalent values (Fig. S1; Table S2). The resulting relationships between 18 GPa and 20 GPa data are adequately described by linear relationships ($R^2 = 0.99$ for all output values sampled, see next paragraph).

Data collected from each species model include total model volume ($mm^3$), total input muscle force (including temporalis, masseter, and medial pterygoid muscle forces, in Newtons), mechanical efficiency (bite nodal restraint reaction force/total muscle input force) at the canine and fourth premolar teeth, respectively, and overall skull strain energy (a measure of work done to deform the skull during simulated bites, in Joules) in canine bite and fourth premolar bite scenarios, respectively (Table 1). All FE simulations portrayed unilateral bites using homogeneous and isotropic material property models, solved by linear static analysis. For a more detailed explanation of the FE modeling workflow see reviews by *Ross (2005)* and *Rayfield (2007)*. For carnivorans and mammals in general, bite force (the magnitude of reaction forces that can be generated at the tooth-food interface, often measured in Newtons) and the related measure of mechanical efficiency (ME; the relative amount of bite force generated per unit of input muscle force,

a unit-less ratio) are important biomechanical traits that broadly correlate with feeding ecology (*Wroe, McHenry & Thomason, 2005*). Strain energy (SE), the amount of work done in deforming a structure (as in deformation of the skull during biting), is thought to be a variable that represents how "energy efficient" a biological structure is in converting input force towards output force rather than towards deforming itself (*Dumont, Grosse & Slater, 2009*). The combination of these two measures of biomechanical performance was previously found to correlate with omnivory (gradual decrease in SE with increasing ME from the anterior to the posterior toothrow) and dietary specialization (presence of noticeable drops in SE in specialized tooth positions as ME increases) (*Tseng & Flynn, 2015*). These observations were subsequently summarized by *Pérez-Ramos et al. (2020)* as relatively SE-invariant increase of ME from anterior to posterior dentition in generalists vs the SE-varying changes in ME for dietary specialists adapted to using specific tooth loci for feeding tasks.

For the reasons outlined above, in this study I focus on ME and SE values of the canine and fourth premolar tooth loci as biomechanical traits that are expected to correlate with dietary breadth (range of food items and therefore food mechanical properties consumed) and trophic level (herbivore and carnivores being more specialized than omnivores). In the extracted dataset, strain energy values were corrected for model volume and input muscle force area differences according to the equation provided by *Dumont, Grosse & Slater (2009)*. The first model in the alphabetically arranged dataset, *Ailuropoda melanoleuca*, was arbitrarily chosen as the standard model to which all other model strain energy values are adjusted to. The final set of biomechanical characteristics calculated for all 27 taxa in the full dataset included Adjusted Canine mechanical efficiency (ACME), Adjusted fourth premolar mechanical efficiency (AP4ME), adjusted strain energy value in canine bite scenario (ADJCSE), and Adjusted strain energy value in fourth premolar bite scenario (ADJP4SE). These data served as the functional performance variables used to characterize a given taxon (Table 1).

The feeding ecological variables used for correlation to the performance variables include dietary breadth and trophic level. The definition of the levels in each variable and the coding for each of the taxon included in the dataset are taken from the PanTHERIA database (*Jones et al., 2009*) (Table 2). Feeding ecological grouping in this study is characterized by the combination of these two categories.

To formalize the qualitative association between performance variables and feeding ecological categories employed in many published studies, I used hierarchical clustering to group the taxa. One dendrogram each was calculated for the FE outputs (performance variables) and dietary ecological traits (dietary breadth and trophic level). The former, continuous multivariate dataset was clustered using Ward's distance measure on an Euclidean distance matrix calculated from the four FE output variables. The latter, categorical bivariate dataset was clustered also using Ward's distance measure, but on a Gower's distance matrix for discrete variables (*Gower, 1971*).

A non-parametric Baker's Gamma correlation coefficient (*Baker, 1974*) was calculated from the two resulting cluster dendrograms to establish the degree of association between the performance variable groupings and dietary ecology groupings of the full dataset.

**Table 2 Feeding ecological variable definitions from the PanTHERIA database (*Jones et al., 2009*).**

| Ecological variable | Definition (from PanTHERIA database) | Value range |
|---|---|---|
| Diet breadth | Number of dietary categories eaten by each species. Categories were defined as vertebrate, invertebrate, fruit, flowers/nectar/pollen, leaves/branches/bark, seeds, grass and roots/tubers | 1 (dietary specialist) to 6 (dietary generalist) |
| Trophic level | Trophic level of each species: (1) herbivore (not vertebrate and/or invertebrate), (2) omnivore (vertebrate and/or invertebrate plus any of the other categories) and (3) carnivore (vertebrate and/or invertebrate only | (1) herbivore (2) omnivore (3) carnivore |

Baker's Gamma counts the level (designated by $k$, the number of clusters) on the dendrogram at which a given pair of taxa is grouped together in both dendrograms, followed by a Spearman (non-parametric) correlation coefficient calculation. This index relies only on the topology of the dendrograms, but not their distance or branch lengths. To assess the influence of branch length information on the resulting correlation measurement, the analysis was repeated using the cophenetic correlation coefficient instead of Baker's Gamma (*Sokal & Rohlf, 1962*). The 95% confidence interval around the calculated correlation coefficient was estimated using bootstrap resampling by simulating a sample of 1,000 dendrograms each in the performance and dietary ecology datasets with randomly assigned tip names from the shared 27-taxon name list. Baker's Gamma and cophenetic correlation coefficients were calculated for each pair of the 1,000 simulated performance and ecology dendrograms. A 95% confidence interval was then calculated from those correlation coefficients.

Next, a series of correlation coefficients between performance and ecology dendrograms were calculated in incrementally smaller sub-datasets of the full dataset of 27 taxa. 1,000 bootstrap samples were generated for each of 24 sets of bootstrap samples, from 26 taxa (1 fewer taxon than full dataset) to 3 taxa (minimum to polarize pairwise comparisons) per dataset. The bootstrapped datasets were pulled from the full dataset with replacement. Median, quantile ranges, mean, and 95% confidence intervals of the mean were calculated for bootstrap replicates at each resampled dataset size. Statistically significant differences were assessed by visually inspecting the 95% confidence intervals of each subsampled dataset against the 95% confidence interval of the full dataset estimated from bootstrap as described in the preceding paragraph. Lack of overlap of 95% confidence intervals indicates a significant difference at the $p = 0.05$ level.

In addition to comparing correlation coefficients between taxon groupings generated from hierarchical clustering of finite element simulation outputs and ecological categorization, respectively, the functional and ecological cluster association to phylogenetic grouping was also assessed within the same analytical framework. A phylogeny with branch lengths based on molecular data was generated from timetree.org for the 27 taxa included in the FE and ecological data comparison (*Kumar et al., 2017*). Branch lengths (in million years) from the phylogeny were used in cophenetic correlation analyses. The phylogeny was then treated as a dendrogram with topology and no branch length information and subjected to correlation analysis using Baker's Gamma coefficient against the FE and ecological datasets, respectively.

It is important to note that the analyses described above make several assumptions about the nature of the data: (1) that the FE simulation outcomes are known without error, (2) that the ecological variables examined are representative of feeding ecology, and that (3) the phylogenetic topology used is accurate and without error. Extensive literature examining the issues behind each of these assumptions demonstrates the complexity of each of these issues in comparative analysis (e.g., *Strait et al., 2005*; *Heath, Hedtke & Hillis, 2008*; *Jones et al., 2009*), but for the sake of the performance-ecology trait comparison focus of this study, those factors were held constant. Furthermore, in addition to the two biomechanical outputs (ME and SE) analyzed in this study, FE simulations produce a plethora of numerical data that can be used to characterize different aspects of structural performance such as stress and strain distributions and magnitudes (*Rayfield, 2007*; *Bright, 2014*). A similarly broad array of ecological and life history traits are available for correlation to biomechanical performance, depending on the research question asked (*Jones et al., 2009*). Recognizing the diverse possibilities for applying FE analyses to study comparative biomechanics, analyses presented herein are intended to highlight the understudied issue of taxonomic sample size in comparative FE analyses using a specific case study of two skull biomechanical traits and two feeding ecological traits in carnivoran mammals.

All data used in the analyses described above are available as Supplemental Files, including the full R script for running all cluster and bootstrap analyses.

## RESULTS

Baker's Gamma correlation coefficient between FE-based taxon clusters and feeding ecology-based taxon clusters in the full data set is 0.0498 (bootstrapped 95% CI [0.0019–0.1260]), indicating weak to no association between taxon groupings generated by FE traits vs ecological traits (Fig. 2A). Of the bootstrapped subsamples, datasets with 12 or more taxa returned correlation coefficients within the range calculated for the full dataset. Datasets with 11 taxa or fewer exhibit increasingly large correlation coefficients as sample size decreased. All taxonomic sample sizes at $n = 25$ or smaller contain at least 1 replicate with correlation coefficient above 0.25, and all taxonomic sample sizes at 13 or fewer contain replicates with correlation coefficients of 0.75 or more.

Baker's Gamma correlation coefficient between FE-based clusters and phylogenetic grouping in the full dataset is 0.0171 (bootstrapped 95% CI [0.0033–0.2451494]), indicating weak to no association between FE traits and phylogenetic structure (Fig. 2B). Resampled taxonomic datasets with 7 or more taxa returned correlation coefficient values within the range observed in the full dataset. Datasets with 6 or fewer taxa exhibited increasingly large correlation coefficients. Resampled datasets with 22 or fewer taxa contained at least one replicate with correlation coefficient above 0.25; datasets with 10 or fewer taxa contained at least one replicate with correlation coefficient value above 0.75.

Baker's Gamma correlation coefficient between feeding ecological clusters and phylogenetic grouping in the full dataset is 0.0498 (bootstrapped 95% CI [0.0015–0.1037]), indicating weak to no association between ecological groupings and phylogenetic groupings (Fig. 2C). Resampled datasets with 12 or more taxa returned correlation

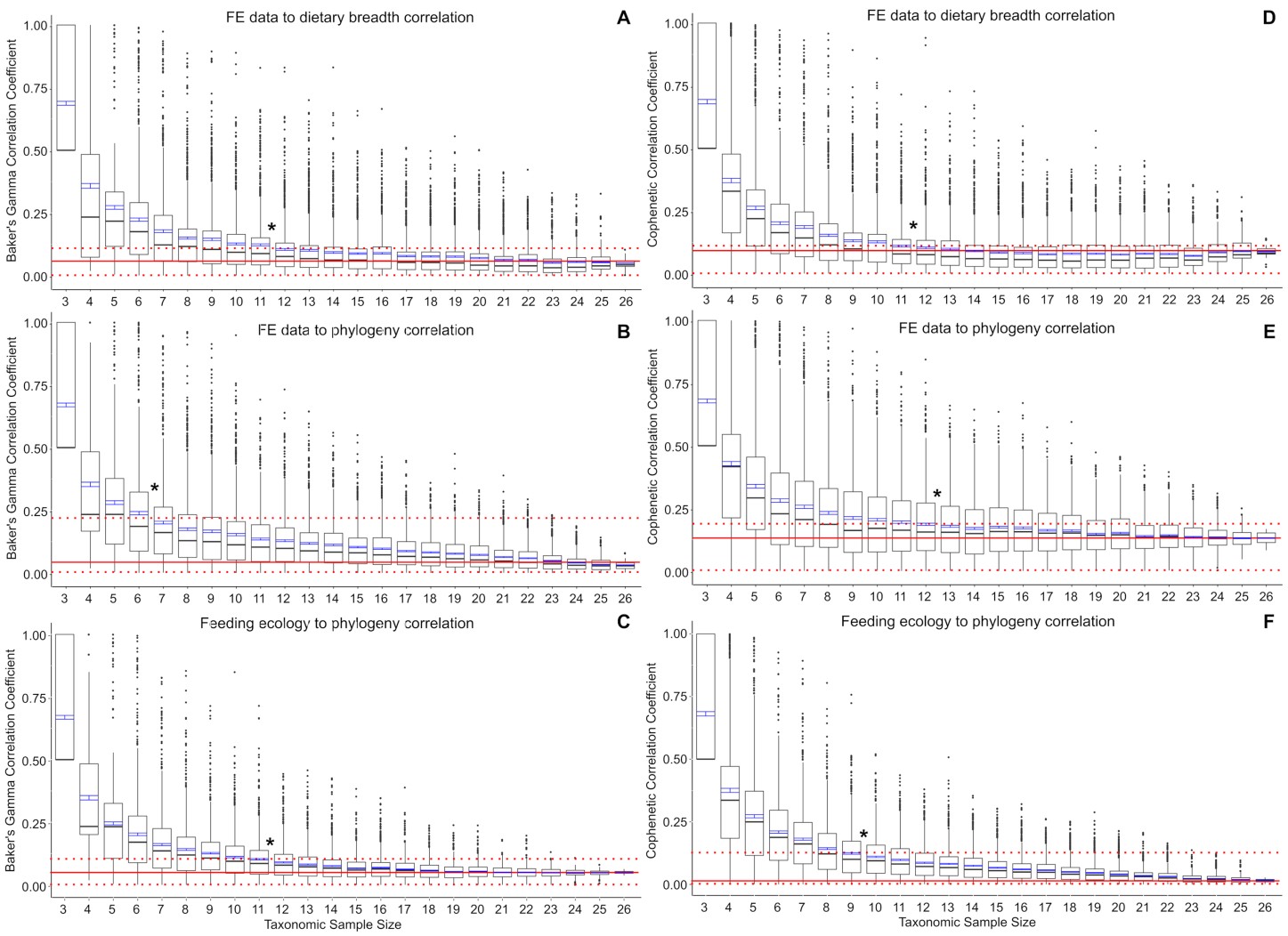

**Figure 2 Correlation coefficients calculated in bootstrap analyses of subsampled datasets.** (A) FE data vs feeding ecology using Baker's Gamma, (B) FE data vs phylogeny using Baker's Gamma, (C) feeding ecology vs phylogeny using Baker's Gamma, (D) FE data vs feeding ecology using cophenetic correlation, (E) FE data vs phylogeny using cophenetic correlation, (F) feeding ecology vs phylogeny using cophenetic correlation. Red solid line indicates correlated coefficient value in the full dataset, dotted red lines represent 95% confidence intervals. Blue bars represent 95% confidence intervals of mean correlation coefficient values at each taxonomic sample size. Boxplots show median values and interquartile ranges. Asterisks indicate sample size above which subsample and full dataset produce similar correlation coefficient values on average.

coefficient values within the range observed in the full dataset. Datasets with 17 or fewer taxa contain at least one replicate with correlation coefficient larger than 0.25; datasets with 8 taxa or fewer contain at least one replicate with correlation coefficient larger than 0.75.

Cophenetic correlation coefficients that consider branch length information from the dendrograms returned broadly similar results to Baker's Gamma correlation coefficient analyses (Figs. 2D–2F). The main differences are found in (1) the FE to phylogenetic structure correlation, which showed that resampled datasets with 13 taxa and above returned correlations not significantly different from those estimated in the full dataset

(compared to 7 taxa and above using Baker's Gamma coefficient), and (2) feeding ecological to phylogenetic structure correlation, which showed that resampled dataset with 10 taxa and above returned correlations similar to those in the full dataset (compared to 12 taxa and above using Baker's Gamma coefficient).

## DISCUSSION

The use of small (<10 taxa) comparative FE model datasets in the majority of published 3D FEA studies of vertebrate skulls likely results in overestimates of the feeding ecological association of simulated biomechanical traits. Relative to a weak or no correlation "full" dataset of 27 taxa, resampled datasets containing 11 or fewer taxa exhibit significantly elevated correlation coefficient estimates for the relationship between FE value and ecological groupings (Fig. 2). Small taxonomic datasets also exhibit stronger association with phylogenetic groupings in both ecological and FE-based clusters, albeit to different extents. Ecological groupings up to ~10-taxon datasets show significantly higher correlation to phylogenetic groupings, and FE-value based groupings show significantly higher correlation to phylogenetic groupings at between 5 and 12 taxa or fewer (Fig. 2). Therefore, at small taxonomic sample sizes of 3–5 taxa, there is relatively high association to phylogenetic structure in the FE data groupings compared to larger taxonomic samples.

The elevated correlation coefficients in smaller taxonomic samples are in part driven by the prevalence of outliers in the bootstrap replicates. The smaller the taxonomic sample, the higher the quantity of high correlation coefficient replicates. This observation suggests taxon sampling choice could have a substantial effect on the resulting presence/absence of significant FE data to ecological grouping correlations. At the low end of the sampling spectrum, all three-sample datasets are expected to return correlation coefficients of 0.5 or higher, with a coefficient of 1.0 defining the upper quartile. This suggests that the practice of using a small number of taxa to perform FE simulations to interpret the overall performance-ecology association of a larger taxonomic clade runs the risk of finding spurious high correlation results when the underlying full dataset exhibits weak or no correlation. In other words, the high number of outliers and the elevated mean correlation coefficients at taxonomic sample sizes smaller than ~11–12 taxa translate to higher incidences of false positives relative to the full dataset.

Despite the inability for small taxonomic datasets to replicate performance-ecology correlations and phylogenetic structure of larger datasets, these findings do not render small taxonomic sample FE studies obsolete. However, the results do highlight the importance of "calibrating" the research question at hand to the appropriate taxonomic sample to lessen potential biases from false positives. Results from small taxonomic sample FE studies could remain useful if the research question is focused on taxon-specific performance-ecology comparisons, rather than extrapolation to broader clade level correlations. Nevertheless, the simultaneous elevated correlation in performance-ecology, performance-phylogeny, and ecology-phylogeny relationships invites caution in interpreting functional correlation when a phylogenetic one is equally plausible based on available data.

To further assess whether the significant correlation of FE and feeding ecological trait data with phylogenetic structure represents elevated phylogenetic signal at smaller taxonomic samples, I conducted post-hoc analyses to estimate phylogenetic signal in the FE and feeding ecology data. I used a multivariate implementation of Blomberg's K (*Adams, 2014*) to calculate phylogenetic signal in FE values in the full dataset as well as a similar bootstrap series of smaller resampled datasets (Fig. 3A). Results indicate that there is a concomitant increase in K with decreasing taxonomic sample size, as has been previously observed for simulated datasets (*Münkemüller et al., 2012*). Increasing K and increasing FE data-feeding ecology correlation with smaller taxon samples suggest the presence of confounding ecological and phylogenetic factors.

I also estimated phylogenetic signal in the categorical ecological data using the delta statistic (*Borges et al., 2019*). In contrast to the elevated phylogenetic signal in FE data at small taxon sample sizes, no elevations in phylogenetic signal are observed in either diet breadth or trophic level categorical data (Figs. 3B and 3C). However, the delta statistic is known to exhibit low sensitivity in detecting phylogenetic signal at small taxon sample sizes (<20) (*Borges et al., 2019*), so it is unclear whether the feeding ecology-phylogenetic structure correlation (Figs. 2C and 2F) is unrelated to phylogenetic signal or whether phylogenetic signal is undetected by current methods. Comparative methods have been shown to be affected by high variance and low power in estimating phylogenetic signal and other parameters at small taxon samples (*Boettiger, Coop & Ralph, 2012*); the elevated FE data, ecology, and phylogeny correlations in the bootstrapped samples of the current study (Fig. 2) appear to be similarly affected by uncertainties at small taxon sample sizes.

Although the bootstrap analysis involves random resampling and it is not possible to pinpoint which specific taxa or phylogenetic factors produce outsized effects on FE-ecology correlations at small taxonomic sample sizes, some general recommendations for small-sample comparative FE studies can still be made. The categorical groupings in the ecological trait data were converted into a continuous distance framework in the cluster analyses via Gower distance measures (see "Methods" section). The result is a relatively clumped dendrogram with polytomies representing ecologically 'identical' taxa in the context of the input variables (diet breadth and trophic level). The overall dataset ($n = 27$) contains multiple samples in each ecological grouping or clump, providing more evenly represented samples in each group than is possible in smaller randomized resampled datasets. As such, a recommended sampling strategy for smaller taxonomic datasets might be to focus on maximizing the even sampling of taxa representing unique ecological trait combinations and to avoid asymmetric sampling of some ecologically similar taxa over others. Such a sampling strategy should reduce spurious high-correlation outcomes when using distance-based comparison methods to study FE-ecology relationships in smaller datasets.

For comparative analyses using FE data to study broader clade-level associations between biomechanical and ecological traits, the time-consuming nature of constructing 3D FE models remains a bottleneck to achieve comparable scope to other data sources such as sequence or geometric morphometric shape data. Given the sensitivity of the

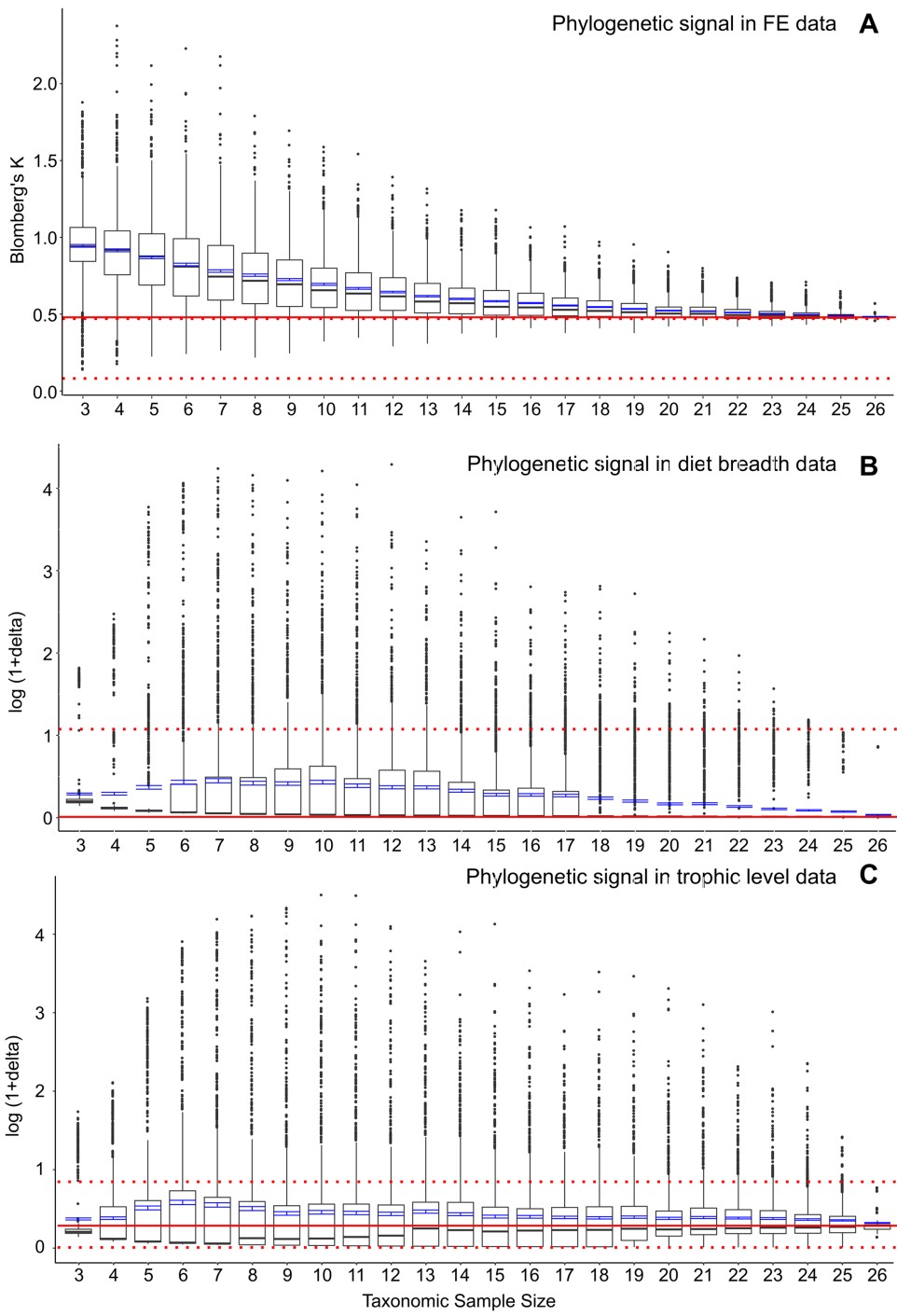

**Figure 3 Phylogenetic signal in bootstrap analyses of subsampled datasets.** (A) FE data, (B) diet breadth data, and (C) trophic level data. Red solid line indicates correlated coefficient value in the full dataset, dotted red lines represent 95% confidence intervals. Blue bars represent 95% confidence intervals of mean correlation coefficient values at each taxonomic sample size. Boxplots show median values and interquartile ranges.

correlation coefficient between FE values and ecological categories on choice in taxonomic sampling and sample sizes, I suggest some rethinking in future comparative FE-based biomechanics research design. Rather than focusing on collection of comparative FE data in increasingly large samples of taxa, I posit that most comparative biomechanics research using the FE method would be better served with a theoretical morphology approach (*Polly et al., 2016*).

The fusion of morphospace analysis (using methods such as morphometric morphometrics or other multivariate trait data) and FE analysis has already been demonstrated to be a fruitful approach to test hypotheses about evolutionary optimality and the relative importance of multiple selective forces in explaining morphological disparity (*Stayton, 2009*; *Tseng, 2013*; *Dumont et al., 2014*; *Polly et al., 2016*). Theoretical morphological models constructed at extremes and/or regular intervals over a given morphospace reduces the subjective nature of taxon selection during FE analyses by summarizing the range of morphological variation that underlies the subsequent FE simulation outcomes, rather than relying on taxonomic-specific interpretations that may be more sensitive to outlier and sampling effects. In this morphospace-driven context, the biomechanical performance context of the taxonomic dataset at hand is only indirectly dependent on the choice of taxon sampling, assuming morphospace sampling is representative of the morphological disparity in the clade studied. A shift to comparative FE analyses based on a morphospace framework leverages the comparative power of the method, especially when the practice of experimental model validation has yet to become standard practice to permit the evaluation of absolute magnitudes of FE simulation outcomes (e.g., *Strait et al., 2005*; *Bright & Rayfield, 2011*).

## CONCLUSION

Time as a limiting factor in applications of 3D FE simulations in comparative biomechanics research has a direct effect on limiting the taxonomic breadth of biomechanics research using comparative FE analysis. A consequence of this limitation is the presence of significant biases in performance-ecology correlation coefficients driven by small taxonomic sample sizes and outliers. Future advances in comparative biomechanics research using FE modeling may depend on a bifurcation in the application of this method. On the one hand, small taxonomic sample studies can remain useful for interpreting taxon-specific biomechanical adaptations, if carefully designed with consideration of phylogenetic structure, ecological trait representation among taxa, and preferably integrated with model validation. On the other hand, research efforts in quantifying clade-level form-function associations could be better served through theoretical morphological approaches of representing morphological disparity, rather than building increasingly large FE datasets of taxon-specific values that are more vulnerable to sampling outlier effects even at larger sample sizes. Continued improvements in model construction efficiency and accuracy are key to solving the time bottleneck issue in using FE methods in broad comparative studies. Finite element analysis is a once *chic* biomechanical modeling method in comparative biology that has come of age, and

continued methodological and application development should help to build its analytical rigor to be on par with any other comparative methodology.

## ACKNOWLEDGEMENTS

I thank J. Liu (UC Berkeley) and members of the Functional Anatomy and Vertebrate Evolution (FAVE) Laboratory for insightful discussions on the ideas and analyses included in this manuscript. J. Luo and J. Marston assisted with model construction. E. Dumont, two anonymous reviewers, and editor J. Hutchinson provided wide-ranging constructive comments that greatly improved all aspects of the study. All errors and insufficiency remain my own.

### Funding

This work was supported in part by the National Science Foundation (U.S.) grant DEB-1257572. The funders had no role in study design, data collection and analysis, decision to publish, or preparation of the manuscript.

### Grant Disclosures

The following grant information was disclosed by the authors:
National Science Foundation (U.S.): DEB-1257572.

### Competing Interests

The author declares that his has no competing interests.

### Author Contributions

- Z. Jack Tseng conceived and designed the experiments, performed the experiments, analyzed the data, prepared figures and/or tables, authored or reviewed drafts of the paper, and approved the final draft.

### Data Availability

Data and scripts are available in the Supplemental Files.

### Supplemental Information

Supplemental information for this article can be found online at http://dx.doi.org/10.7717/peerj.11178#supplemental-information.

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
