# Peer review of "Rethinking the use of finite element simulations in comparative biomechanics research"

_PeerJ, doi:10.7717/peerj.11178_

## Round 0.1 · original submission · Major Revisions

Three reviewers have given very thoughtful input on your MS, and agree that it is a useful contribution to the literature. Their constructive critiques range widely so I will not attempt to summarize them here, but please address all points individually in your Response. Thank you!

Reviewer 1 ·

Basic reporting

This manuscript is very-well written, with no grammatical typos and clear, easy-to-follow language. It is structured in a standard, professional manner and crucial data is contained within the supplementary data. As the article now stands, the relevant background literature is cited, although some additional references may be needed - please see my comments below.

Overall goal of the manuscript is clearly stated in the paragraph starting Line 47.

The study is mostly self-contained although some important data (included in the two previous studies from which data for the meta-analysis are taken) should be included here, please see further comments.

A few minor suggestions for grammatical changes are made in the attached annotated PDF.

Experimental design

As noted in Part 1, the overall research question and how this study fills an important knowledge gaps are clearly stated in the Introduction.

Generally, the methodology applied is sound. The author makes use of finite element models from two previous studies on which they were an author (Prybyla et al 2018 and Perez-Ramos et al 2020). The selection of these models is well-supported - they were analysed using the same FE software (Strand7), muscle forces were calculated using the dry skull method, applied with Boneload and similar percentages used for the working and balancing sides, similar constraints were applied and the selected output (ME and SE) were scaled to account for differences in sizes. One minor thing missing is material properties applied, as the elastic moduli used in the two studies varied slightly (18 vs 20 GPa) - please specify what was used here.

Furthermore, the author is to be commended for clearly discussing assumptions that meta-analyses make of the data (Line 153).

However, there is some information missing in the methods section. First, Prybyla et al 2018 included 21 taxa and Perez-Ramos et al 2020 included 12 taxa yet the current study makes use of 27 taxa - why were the others excluded? Although full information on the taxa used is in the supplementary table, this should be noted, as well as whether the taxa analysed here were extinct or fossil (both were analysed in the earlier studies).

In the supplementary table "eco-dataset" there is no key for the number coding for dietary breadth or trophic level. Please include some key with this supplementary table - do not make readers have to look this up in PanTHERIA.

I definitely feel that the crucial information contained within the supplementary table "FE dataset" is important enough to be a table within the main text, rather than supplementary data.

My biggest concern with experimental design and interpretation of results involves the output variables/performance metrics considered. Both are measures of structural efficiency (in terms of conversion of muscle force to bite force and strain energy) that are then associated to ecological variables and phylogeny. There are two points to be made here:

1. It is not explained in the text HOW these variables are expected to vary with (and therefore help identify) different diets or ecologies. Why use these metrics? Do we expect herbivores to have more efficient skulls than carnivores, for example? I realise the author covered this to some extent in a previous paper, Tseng and Flynn 2015, but a brief explanation is needed here. Additionally, why were two bite points tested? What do differences in results between these bite points signify? Again, in Perez-Ramos et al 2020 it is stated that larger differences in results between the two bite points suggests higher functional differentiation along the tooth row and therefore more restrictive diets. Such explanations - even briefly - would be useful here, otherwise it is unclear why these metrics are used.

2. One of the wonderful and tricky things about FEA is the sheer amount and diversity of data that is produced. Most FE studies only consider a small subset of output to analyse, usually targeted to the hypothesis being tested. In this case, the author chooses two measures of efficiency. However, it is very important to note that efficiency is not the only performance metric linked to ecology and diet. For example, some taxa with inefficient jaw systems (for example, crocodilians) still produce extremely high (relative to body size) bite forces which strongly influence ecology. Other important outputs include peak stress/strain magnitudes - I will note that great care must be taken when extracting and interpreting such values as validation studies have shown FEA is not 100% accurate in predicting such values compared to experimental data. Nonetheless, relative comparisons of peak stress/strain between scaled models is an important indicator of structural strength. Also important are more qualitative outputs such as von Mises stress distribution (areas of high vs low stress) and how the skull deforms under loads (torsion, bending, etc.).

I am certainly NOT saying that the author should consider all of these performance metrics in the current study, which would make it substantially longer and more complicated. However, I think it must be noted somewhere in the main text that other outputs strongly linked with diet and ecology are not being analysed and that this study only considers a small (but important) subset of performance metrics available.

A few additional and minor notes. How does the quite unequal distribution of dietary breadth categories (for example, one each in types 4 and 5 and none in type 2) and trophic level (predominantly type 2, with few from type 1) impact statistical analyses?

Given that Fig 1A is used to illustrate the increased application of FEA in biosciences and that the present analysis focuses on the skull, the first paragraph of the Survey Methodology could be substantially trimmed. Similarly, it would be useful to know the number of studies focused on the skull illustrated in Fig 1B.

Validity of the findings

Overall the results of this study are very interesting and I think will prove useful in designing future studies using FEA. The main point is that smaller data sets produce stronger correlations between FE results and ecology/phylogeny than increasingly larger datasets. It will be interesting to see whether this trend holds true for other clades/anatomical regions/biomechanical techniques/FE performance metrics. Nonetheless, the results presented here are intriguing.

The author's comments on the importance of taxon sampling - and in particular the influence of outliers - are particularly fascinating. One question - do you have any evidence that demonstrates that the smaller samples tended to include a larger proportion of outliers? If so, are there any recommendations to identify prior to designing a study which taxa are outliers and may impact results (morphometrics?)?

I agree with the conclusion of the need to more carefully design FE studies by considering the taxa and metrics needed to address specific questions. Also interesting is the suggestion made that one possible solution is the use of theoretical modelling to eliminate outliers by providing a morphological/functional continuum.

A minor note - the opening lines of the first three Results paragraphs imply weak to no correlation between FE results and ecological traits/phylogeny for the full data set. Yet both previous studies from which these models are derived (Prybyla et al 2018 and Perez-Ramos et al 2020) did draw substantial correlations between these results and diet in fossil taxa (the sample sizes in those studies were smaller than the full data set here but still above the thresholds identified in the current study). Why did the same models produce more meaningful results in the previous studies but not here?

Additional comments

Figure 1: The X axis in part B would be easier to understand using ranges, i.e., 1 – 4, 4 – 7, etc. rather than the current notation using parentheses and brackets.

Annotated reviews are not available for download in order to protect the identity of reviewers who chose to remain anonymous.

·

Basic reporting

The manuscript is succinct with a clear and easy flow from Introduction through Conclusions.

The literature cited is appropriate and sufficient for the purposes of this study. The paper does a good job of articulating the primary issues and approaches associated with the use of FE analysis in a comparative framework.

The FE variables in the data file do not match the variables in the text. Changing CME and P4ME in the text to ACME and P4ME will solve that issue. For ease of reading, consider adding captions to Figure 2 (e.g., along the lines of, FEA results and feeding, FEA results and phylogeny, Feeding and phylogeny).

Experimental design

The approach of correlating dendrograms is clever and accompanied by appropriate implementation of bootstrapped distributions.

The methods are clearly described and can be replicated with the data and code provided. The only minor clarification that would be helpful is whether the phylogeny dendrogram included branch lengths. Branch lengths in the feeding and FEA dendrograms were equal. If the phylogeny dendrogram includes branch lengths, then consider adding a comment about the impact, or lack thereof, of correlating dendrograms with equal and unequal branch lengths.

Consider providing a brief description of how dietary breadth and trophic level are defined in PanTHERIA. Opinions about categorizing feeding ecology vary widely and briefly addressing the method will help readers assess their level of comfort with the data.

The list of papers included in the meta-analysis may be a useful resource for future studies. Consider including those data.

Validity of the findings

The results are directly relevant to the hypotheses and stand alone (and beg several interesting questions).

The study confirms that small sample size can lead to elevated correlations between feeding ecology and FE-based performance measures relative to the clade-level pattern.

Additional comments

The suggestion that clade-level analyses of FE-based performance variables focus on “theoretical morphological approaches of representing morphological disparity” is thoughtful and forward-thinking. As noted, several studies have already provided approaches to mapping FEA performance variables into theoretical morphospace. Some have concluded that there is no evidence of selection for FEA-based variables that were hypothesized to reflect performance in the early days of comparative FEA studies. From that perspective, some comparative FEA analyses are already rising to meet the challenge of scrutiny and analytical rigor that is called for in the concluding sentence. Meanwhile, most recent studies, even those with small sample sizes, acknowledge the need for broader taxonomic sampling refrain from claiming to have identified adaptations. There is a great deal of room for innovation in using applying FEA within comparative framework, but perhaps the current state of progress is not as bleak as it seems.

Reviewer 3 ·

Basic reporting

no comment

Experimental design

no comment

Validity of the findings

no comment

Additional comments

This is a clear manuscript that identifies an issue which has yet to really emerge in FE studies, which is how low sample sizes can distort statistics. The reason that this issue hasn’t been identified before is that 1) very few studies have attempted to investigate more than (ballpark) five or so species at a time, and 2) even fewer have attempted to do any kind of quantitative analysis of the results, and mostly stop at visual comparison of contour plots. This is especially true of 3D models (although Marce-Nogue et al (2017: https://doi.org/10.7717/peerj.3793) have put forward some interesting methods to quantitatively compare 2D FE models using principal components analysis).

I think that this manuscript provides a useful warning for future FE studies as they grow in ambition and scope. However, I do have a broad question regarding the low correlations between FE-ecology, FE-phylogeny, and ecology-phylogeny, and why these correlations are so low in the full dataset. To an extent I would expect FE correlations to be low, because the metrics used in the analysis are themselves quite broad (e.g. total strain energy) and thus may not highlight the most ecologically or phylogenetically relevant differences between taxa (for example, one could argue that the total strain is not as important as whether/where that strain is concentrated). But the ecology-phylogeny one is surprising. Presumably there is a real relationship between ecology and phylogeny, and if you were to treat the ecological categories as tip traits and then test for a phylogenetic signal, you would find one? So it’s weird to me that as you add data, this correlation goes down. Can you explain why this is? If you can, does that explanation extend to the low FE correlation results too? If you can’t, does that mean that a cluster analysis isn’t appropriate for this type of data/question (I’ll admit that I don’t know much about cluster analyses, but presumably a cluster is less ‘constrained’ than a phylogeny because it doesn’t care about history? I may not be phrasing that correctly…). If you looked for a more traditional phylogenetic signal in your FE results (e.g. k, or lambda), would you find one that way?

Pushing it right to the edge, if those low correlations are real, what does that mean about our efforts more broadly to understand function from an evolutionary or ecological perspective? I realise these questions are somewhat tangential to your main point, which is that we need to think carefully and critically about what we’re measuring and why (I completely agree with this), but you’ve piqued my curiosity!

Minor adjustments:
L33: “covers”
L69: “The total number … was counted”
L217: The word ‘valid’ has a very specific meaning in FE studies, and implies that the model is producing realistic outputs. While context does indicate that you’re being more colloquial, a less loaded word (‘useful’?) would be more appropriate.

Code:
I’ve had all sorts of issues when packages stopped working because I was using the wrong version of R. I think it’s a good idea to comment which versions of R and the packages you used when you wrote the script, so that people can rollback if they need to.

The packages ‘datelife’ and ‘phylocurve’ don’t appear to be on CRAN at the moment. I was able to find versions of them on Github, but couldn’t install phylocurve because of an Rtools issue that I couldn’t be bothered to fix. Apparently, this doesn’t matter because everything ran without errors, which implies that phylocurve isn’t called anyway? Regardless, I’m not sure if these packages are going to go back on to CRAN at some point, but in the meantime I’d consider putting the Github addresses in for people.

Library function loads ape twice

code L22: .csv file extension missing

---

## Round 0.2 · accepted · Accept

I agree with the reviewer; this is a wonderful contribution to the literature and you've done a fabulous job incorporating the very helpful reviews. Congrats on acceptance!

Reviewer 1 ·

Basic reporting

Again, this is a well-written and structured manuscript with all crucial data and methods as well as relevant background literature included, and study goals clearly stated.

The author has made all suggested minor grammatical changes in this revision.

Experimental design

I commend the author for clarifying what specimens were included (and why) from the two previous studies, including information on dietary breadth and trophic level within the manuscript (rather than referring readers to PanTHERIA) and for their thorough sensitivity analysis of the impact of slight differences in material properties between models taken from the two earlier studies - this addresses my concerns from the original version of the manuscript, thank you!

I am also satisfied with the author addressing the fact that while two key performance metrics (ME and SE) are analysed in this study, finite element modelling produces a diverse and enormous range of outputs that are beyond the aims of the study here, but nonetheless are very important. I also thank the reviewer for clearly describing why the two chosen metrics are important and what these outputs mean in terms of differentiating diet and ecology.

Validity of the findings

Again, the findings of this study are very interesting and will prove very useful in designing future FEA studies. Many thanks to the author for expanding the discussion on the impact of outliers on smaller sample sizes and their explanation for why more substantial correlations between FE outputs and diet in previous studies from which models were derived.

Additional comments

Once again, thank you for your substantial work on this revised version and your very clear point-by-point discussion of all issues raised in your rebuttal letter. I feel this manuscript is now ready for publication and look forward to seeing it in print!